# Key Amino Acids for Transferase Activity of GDSL Lipases

**DOI:** 10.3390/ijms232315141

**Published:** 2022-12-01

**Authors:** Takanori Yamashiro, Akira Shiraishi, Koji Nakayama, Honoo Satake

**Affiliations:** 1Dainihon Jochugiku Co., Ltd., 1-1-11 Daikoku-cho, Toyonaka 561-0827, Osaka, Japan; 2Department of Chemical Science and Engineering, Graduate School of Engineering, Kobe University, 1-1 Rokkodai-cho, Nada-ku, Kobe 657-8501, Hyogo, Japan; 3Bioorganic Research Institute, Suntory Foundation for Life Sciences, 8-1-1 Seikadai, Seika-cho, Souraku 619-0284, Kyoto, Japan

**Keywords:** *Tanacetum cinerariifolium*, GDSL lipase, transferase, in silico

## Abstract

The Gly-Asp-Ser-Leu (GDSL) motif of esterase/lipase family proteins (GELPs) generally exhibit esterase activity, whereas transferase activity is markedly preferred in several GELPs, including the *Tanacetum cinerariifolium* GDSL lipase TciGLIP, which is responsible for the biosynthesis of the natural insecticide, pyrethrin I. This transferase activity is due to the substrate affinity regulated by the protein structure and these features are expected to be conserved in transferase activity-exhibiting GELPs (tr-GELPs). In this study, we identified two amino acid residues, [N/R]208 and D484, in GELP sequence alignments as candidate key residues for the transferase activity of tr-GELPs by two-entropy analysis. Molecular phylogenetic analysis demonstrated that each tr-GELP is located in the clusters for non-tr-GELPs, and most GELPs conserve at least one of the two residues. These results suggest that the two conserved residues are required for the acquisition of transferase activity in the GELP family. Furthermore, substrate docking analyses using ColabFold-generated structure models of both natives and each of the two amino acids-mutated TciGLIPs also revealed numerous docking models for the proper access of substrates to the active site, indicating crucial roles of these residues of TciGLIP in its transferase activity. This is the first report on essential residues in tr-GELPs for the transferase activity.

## 1. Introduction

The Gly-Asp-Ser-Leu (GDSL) motif esterase/lipase family proteins (GELPs) are lipases that feature a Gly-Asp-Ser-X (GDSX) consensus motif and are involved in a wide variety of biological functions [1], including seed germination [2], pollen interaction [3], lipid metabolism [4], and secondary metabolism [5]. Canonical GELPs exhibit esterase (hydrolysis) activity, but several GELPs preferentially function as transferases rather than esterases. In *Tanacetum cinerariifolium*, the species-specific insecticidal secondary metabolite pyrethrin I is biosynthesized via the esterification of chrysanthemoyl-CoA and pyrethrolone by the *T. cinerariifolium* GDSL lipase, TciGLIP [6]. Although TciGLIP also exhibits esterase activity against its transferase product pyrethrin I, this esterase activity is much lower than the transferase activity [6]. *Tanacetum coccineum*, a phylogenetically close species to *T. cinerariifolium*, can also produce pyrethrins, and TcoGLIP (*T. coccineum* GDSL lipase) has been found in the genome of *T. coccineum* [7,8]. In other genera, TaXAT (*Triticum aestivum* xanthophyll acyltransferase) catalyzes the esterification of xanthophyll and triacylglycerides to xanthophyll esters, and like TciGLIP, it exhibits much higher transferase activity than esterase activity [9]. Moreover, SlCGT (*Solanum lycopersicum* chlorogenate: glucarate caffeoyltransferase) has been reported as the first GELP family enzyme that has acquired transferase activity and lost its reverse esterase activity [10].

Such transferase activity of GELP family proteins is likely due to substrate affinity caused by the protein structure, and these features are expected to be shared among transferase activity-exhibiting GELPs (tr-GELPs). Many studies on GELPs have thus far shown that the general active site for esterase activity is formed by three amino acid residues, Gly-Asp-His, designated as a “catalytic triad” [11]. Point mutations in the catalytic triad of TciGLIP (S40A, D318A, or H321A) result in a loss of its transferase activity [6,12], whereas a similar mutation in SlCGT (H331A) does not affect its transferase activity [10], indicating that crucial residues for transferase activity do not exactly accord with that for esterase activity. These findings suggest the existence of other crucial residues for the transferase activity of tr-GELPs aside from the catalytic triad. In this study, we identified common key residues for transferase activity in tr-GELPs using a two-entropy analysis of amino acids, and the effects of these residues on TciGLIP were estimated using structure model prediction and docking simulations. To the best of our knowledge, this is the first report on amino acids responsible for the transferase activity of tr-GELPs including TciGLIP.

## 2. Results and Discussion

### 2.1. Sequence Alignment and Two-Entropy Analysis of GELPs

For the GELPs with known substrates, the protein sequences of four tr-GELPs, including TciGLIP [6], TcoGLIP [7], TaXAT [9], and SlCGT [10], and six esterase activity-exhibiting GELPs (est-GELPs), including AtCDEF1 (*Arabidopsis thaliana* cuticle destructing factor 1 [3]), BnSCE3 (*Brassica napus* sinapine esterase [2]), CpEST (*Carica papaya* esterase [13]), FvGELP1 (*Fragaria vesca* GDSL esterase/lipase [14], OsGLIP1 (*Oryza sativa* GDSL lipase [4]), and RsAAE (*Rauvolfia serpentina* acetylajmalan acetylesterase [5]), were collected from the NCBI database. In addition to these GELPs, putative GELPs were detected by BLASTP searches of the NCBI NR database with each of the GELPs as a query. All collected GELP sequences were aligned using CLUSTAL W-mpi [15] (Figure 1).

To detect amino acid residues conserved preferentially in tr-GELPs, two-entropy values of amino acids at each position in the sequence alignments were calculated according to previous studies [16,17]. For instance, the two-entropy analysis has been employed to determine the ligands of adenosine receptors [18] and the ligand recognition mechanism of cannabinoid receptors [19]. The difference in the entropy value between certain protein groups is zero at positions where there is no correlation between the function and amino acid type. Since the entropy values of positions with a high correlation between function and amino acid type are high, the positions with a low entropy of tr-GELP and a high entropy of other GELPs (est-GELPs and putative GELPs) are the residue positions that are crucial for transferase activity. Figure 2 shows scatter plots of positions in the alignment and the distances from the catalytic triad in TciGLIP respectively, demonstrating the difference in entropy values between tr-GELPs and other GELPs. No region-specific difference was detected in entropy values, except for a higher entropy for tr-GELPs at the N-terminal signal sequence. Among the top ten residues that had lower entropy values in all four tr-GELPs compared to other GELPs, Asn or Arg at position 208 ([N/R]208) and Asp at position 484 (D484) in the alignment are not present in the six est-GELPs (Appendix A). In contrast, residues that were preferentially conserved in est-GELPs were not detected. These data indicated that [N/R]208 and D484 are crucial for the acquisition of tr-GELP activity.

### 2.2. Molecular Phylogenetic Analysis of GELPs

A molecular phylogenetic tree of tr-GELPs, est-GELPs, and putative GELPs is shown in Figure 3. In the Asteraceae family (including *T. cinerariifolium* and *T. coccineum*) protein cluster (Figure 3, clade A), only TciGLIP and TcoGLIP GELPs harbor both [N/R]208 and D484, and several GELPs conserved either [N/R]208 or D484. In the Solanaceae family (including *S. lycopersicum*) proteins, GELPs conserving D484 formed a cluster (Figure 3, clade B), and two GELPs with both [N/R]208 and D484 were found beside SlCGT in this cluster. In *Triticum* and *Hordeum* genera plant proteins, TaXAT, six GELPs with both [N/R]208 and D484, and three GELPs with D484 formed a cluster (Figure 3, clade C), suggesting that the six GELPs with both [N/R]208 and D484 exhibit similar transferase activity to TaXAT. In addition to the four known tr-GELPs, nine GELPs with both [N/R]208 and D484 formed a cluster consisting of Brassicaceae proteins (Figure 3, clade D). Interestingly, this cluster is located distantly from other Brassicaceae protein clusters, including est-GELP BnSCE3 (*Brassica napus* sinapine esterase) (Figure 3, clade D-II), suggesting that these GELPs might have been multiplied separately in the same plant family.

### 2.3. Prediction of Protein Structures and Substrate Docking Simulations of TciGLIP

Access of a substrate to an active site is a prerequisite for enzymatic activity. Structural models showing proper access for a substrate to an active site are defined as “reasonable models”, and the correlation between the number of reasonable models and the affinity is employed to predict the structure-function correlations of enzymes [20,21]. To examine the contribution of R153 and D336 to the transferase activity of TciGLIP (corresponding to [N/R]208 and D484 in the sequence alignment, respectively), we assessed the number of models that allowed access of the substrates of pyrethrin I (chrysanthemoyl-CoA and pyrethrolone) to the catalytic triad of native TciGLIP [12], virtual mutants (R153A-TciGLIP and D336A-TciGLIP), and experimentally validated mutants (S339A-TciGLIP and G64A-TciGLIP), since the substitution of S339 with A fails to affect the transferase activity of TciGLIP [6], whereas the substitution of G64 with A results in complete loss of the activity [12]. The amino acid sequences of the native TciGLIP and the four TciGLIP mutants (S339A, G64A, D336A, and R153A) were subjected to ColabFold analyses [22] to predict their protein structures. For each structure model, chrysanthemoyl-CoA and pyrethrolone were docked using AutoDock Vina (Figure 4), and the number of reasonable models was counted.

These analyses generated 45.3 ± 7.2 reasonable models for native TciGLIP (Figure 5A). Likewise, 35.0 ± 4.5 reasonable models were generated for S339A-TciGLIP (Figure 5A), and no significant difference was detected between the two transferase-positive TciGLIP proteins (Figure 5A, *p* < 0.05, Dunnett’s test). In contrast, the number of reasonable models for G64A-TciGLIP (25.0 ± 10.2) was significantly lower (Figure 5A, *p* < 0.05, Dunnett’s test). These results were in good agreement with previous experimental results demonstrating that S339A-TciGLIP has equipotent transferase activity to native TciGLIP [6], whereas G64A-TciGLIP is devoid of transferase activity [12]. Of particular interest is that both the numbers of reasonable models of the D336A mutant (25.3 ± 6.3) and the R153A mutant (21.0 ± 5.4) were significantly smaller than that of native TciGLIP (Figure 5A, *p* < 0.05, Dunnett’s test) and comparable to that of G64A-TciGLIP. These results indicated that both D336 and R153 are crucial for the transferase activity of TciGLIP. Notably, both D336 and R153 are located distantly from the catalytic triad of TciGLIP (Figure 2B and Figure 5B) and not in conserved regions, Blocks I, II, III, or V, among GELPs [6] (Appendix A). Collectively, these structural analyses suggest that R153 and D336 participate in distal regulation of the active confirmation responsible for the transferase activity of TciGLIP.

## 3. Materials and Methods

### 3.1. Sequence Collection and Phylogenetic Analysis

The protein sequences of GELPs with known substrates, including four tr-GELPs (TciGLIP: AFJ04755.1 [6], TcoGLIP: GJR32646.1 [7], TaXAT: QEM23753.1 [9], and SlCGT: CBV37053.1 [10]) and six est-GELPs (AtCDEF1: NP_194743.1 [3], BnSCE3: Q3ZFI4.1 [2], CpEST: P86276.1 [13], FvGELP1: XP_004304671.2 [14], OsGLIP1: APX55003.1 [4], and RsAAE: AAW88320.1 [5]), were collected from the NCBI database. The protein sequences of putative GELPs were detected by BLASTP searches (version 2.13.0, National Center for Biotechnology Information Bethesda, MD, USA) against the NR database with each of the tr-GELPs and est-GELPs as queries. The top 30 BLASTP hit sequences were screened using an E-value cut-off of 10^−3^, and after removing redundant sequences, they were used as putative GELPs. All collected GELPs were aligned using CLUSTAL W-mpi 0.13 [15] with BLOSUM62. A maximum-likelihood phylogenetic tree was created using Fast Tree 2.1.10 [23] based on a JTT matrix-based model [24] with 100 bootstraps.

### 3.2. Two-Entropy Analysis of GELPs

To find amino acid positions from a multiple sequence alignment that correlate with transferase activity, we calculated two of Shannon’s entropy values according to a previous study [16]. In short, the entropy values for position *p* (*E_p_*) for tr-GELPs and other GELPs are given by:$$E_{p}=-\overset{20}{\underset{a=1}{\sum }}\frac{N_{a,p}}{N_{all}}\log_{10}\frac{N_{a,p}}{N_{all}}$$
where *N*_*a*__,*p*_ is the number of sequences with amino acid *a* at alignment position *p* that was corrected using a BLOSUM62-based pseudo-count strategy [17]. *N_all_* represents the number of sequences within the alignment. The pseudo-count was set to 2.00.

### 3.3. Protein Structure Modeling and Substrate-Binding Simulations of TciGLIP

The protein sequences of native TciGLIP and four TciGLIP mutants (S339A, G64A, D336A, and R153A) were subjected to structure model prediction using ColabFold (AlphaFold2 with MMseqs2) [22] with default parameters, and we extracted the five models with the lowest energy for subsequent analyses. The structures of pyrethrin I substrates, chrysanthemoyl-CoA (CHEBI: 143950), and pyrethrolone (CHEBI: 39111), were downloaded from the ChEBI database (https://www.ebi.ac.uk/chebi/init.do (accessed on 16 August 2022)) and minimized with CHARMM force fields using Spartan’18 v1.4.5 (Wavefunction, Inc., Irvine, CA, USA). The top five models of each TciGLIP mutant were subjected to docking modeling with chrysanthemoyl-CoA using AutoDock Vina 1.1.2 [25]. The numbers of grid points in the x-, y-, and z-axes for AutoDock were 36 × 42 × 44, with grid points separated by 0.375 Å; we set exhaustiveness to 100, num_modes to 20 (max), and other parameters to default settings. The chrysanthemoyl-CoA docked models were selected based on the distance between the sulfur atom of the chrysanthemoyl-CoA thiol ester region and the imidazole group of the His321 C-2 atom in the catalytic triad of TciGLIP. Furthermore, pyrethrolone was docked to the chrysanthemoyl-CoA docked models by using AutoDock Vina 1.1.2. The numbers of grid points in the x-, y-, and z-axes for AutoDock were 18 × 24 × 22, with grid points separated by 0.375 Å; we set exhaustiveness to 100, num_modes to 20 (max), and other parameters to default settings. The numbers of reasonable models in which both substrates (chrysanthemoyl-CoA and pyrethrolone) were close to the catalytic triad were counted for every protein sequence model, and these numbers were examined for statistical significance with Dunnett’s test, with *p* < 0.05 indicating significance. Visualization of the protein models was performed using UCSF Chimera 1.16 [26].

## 4. Conclusions

In this study, we have originally demonstrated two candidate key residues for the transferase activity of tr-GELPs by a combination of two-entropy analysis, predictive structure modeling, and docking simulations. The present study paves the way for investigating the evolutionary molecular mechanisms underlying the acquisition of transferase activity. Experimental validation of the functional roles for R153 and D336 on the transferase activity of TciGLIP is underway.

## Figures and Tables

**Figure 1 ijms-23-15141-f001:**
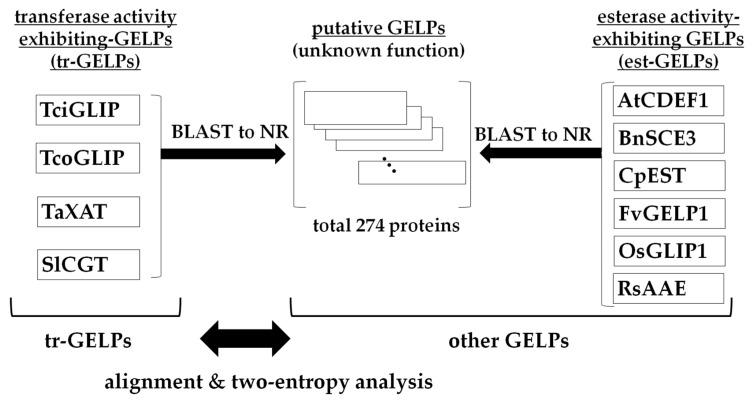
GELPs used in this study. TciGLIP, TcoGLIP, TaXAT, and SlCGT are tr-GELPs. AtCDEF1, BnSCE3, CpEST, FvGELP1, OsGLIP1, and RsAAE are est-GELPs. Putative GELPs were detected using BLASTP searches with each of the tr-GELPs and est-GELPs as queries, though their functions are unknown. Est-GELPs and putative GELPs are collectively called “other GELPs” in this study. All GELP types are underlined.

**Figure 2 ijms-23-15141-f002:**
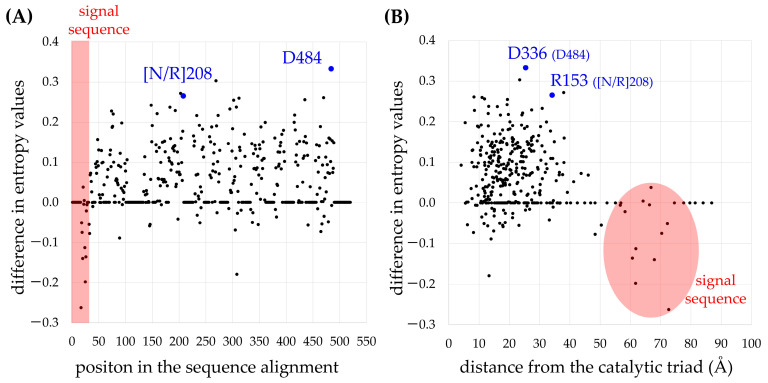
Scatter plots of the difference in entropy values of each amino acid between tr-GELPs and other GELPs (est-GELPs and putative GELPs). The y-axis of both plots is the difference in entropy values that was calculated by subtraction of the entropy value of each amino acid in tr-GELPs from that of other GELPs. A higher y-axis value indicates amino acid residues that were conserved preferentially in tr-GELPs compared to other GELPs. (**A**) The x-axis is the sequence position in the alignment. (**B**) The x-axis is the distance from the catalytic triad of TciGLIP. R153 and D336 in TciGLIP correspond to [N/R]208 and D484 in the sequence alignment (Appendix A), respectively.

**Figure 3 ijms-23-15141-f003:**
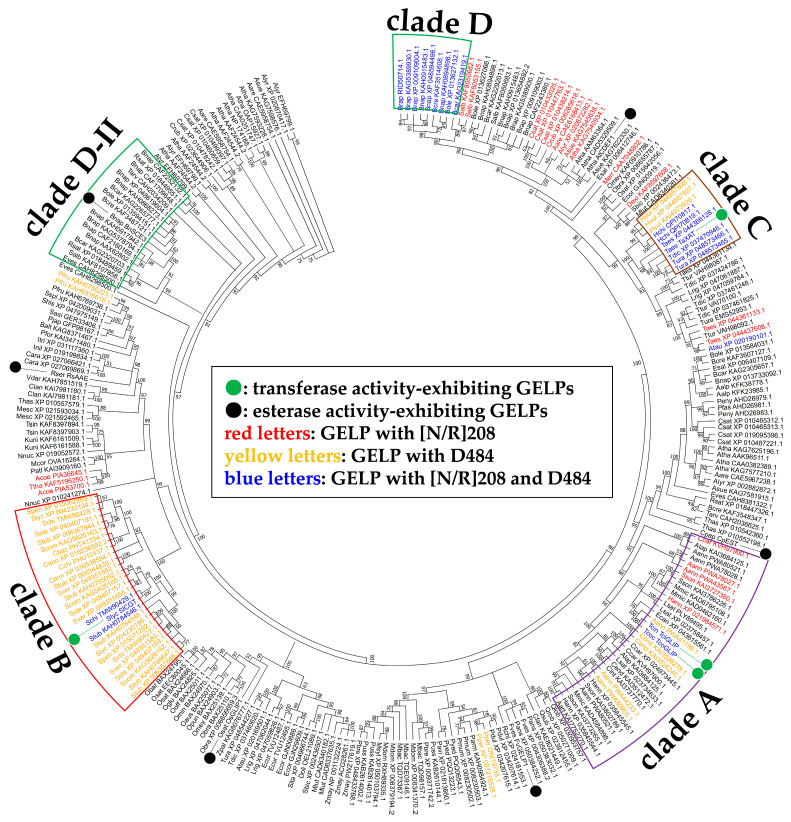
Phylogenetic tree of GELPs analysed in this study. Each protein name is indicated by a species abbreviation (Appendix A) and common name or accession number. Green circles and black circles indicate tr-GELPs and est-GELPs respectively. Red, yellow, and blue letters indicate GELPs with [N/R]208, D484, and both [N/R]208 and D484, respectively. The Asteraceae family (clade A), Solanaceae family (clade B), *Triticum* and *Hordeum* genera, (clade C), and Brassicaceae family plant protein clusters (clades D and D-II) are surrounded by purple, red, brown, and green lines, respectively.

**Figure 4 ijms-23-15141-f004:**
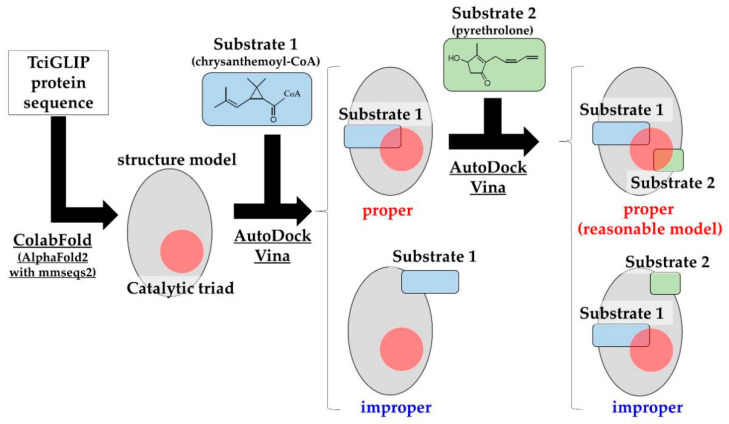
Scheme of selecting reasonable models. The protein sequences of TciGLIPs (native and mutant) were subjected to ColabFold to generate structure models. Chrysanthemoyl-CoA (substrate 1) was docked using AutoDock Vina for each model. Moreover, the simulated models in which substrate 1 was close to the catalytic triad of TciGLIP were subjected to AutoDock Vina with pyrethrolone (substrate 2) as with substrate 1. Simulated models in which both substrates 1 and 2 are close to the catalytic triad of TciGLIP are regarded as “reasonable models”. All pieces of software used are underlined.

**Figure 5 ijms-23-15141-f005:**
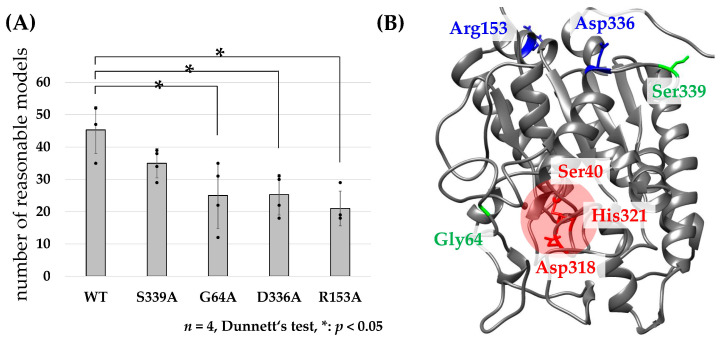
(**A**) The numbers of reasonable models of TciGLIPs. (**B**) Predicted structure of native TciGLIP and characteristic residues shown by UCSF Chimera 1.16. Red letters (Ser40, Asp318, and His321) indicate the catalytic triad of TciGLIP. Green letters (Gly64 and Ser339) indicate residues that have been experimentally validated by mutagenesis in previous studies. Blue letters (Arg153 and Asp336) indicate candidate key residues for transferase activity as detected in this study.

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
