# Peer review of "Key Amino Acids for Transferase Activity of GDSL Lipases"

_ijms, 2022, doi:10.3390/ijms232315141_

Round 1
Reviewer 1 Report
1. Overall well-written article.
2. Somewhere in "2. Results and Discussion" the author should conclude their finding and suggest some future direction for this study based on current findings.
Author Response
<Response>
We thank this reviewer for evaluation and helpful comment. As per your comments, we moved the conclusion of the results “section 2.3” into “Section 4. Conclusions” to clarify our conclusion. We have added “Section 4. Conclusions” as follows:
In the Conclusions section
“In this study, we have originally demonstrated two candidate key residues for transferase activity of tr-GELPs by a combination of two-entropy analysis, predictive structure modeling, and docking simulations. The present study paves the way for investigating the evolutionary molecular mechanisms underlying the acquisition of transferase activity. Experimental validation of the functional roles for R153 and D336 on the transferase activity of TciGLIP is underway.” (Line 226 in the revised manuscript)
Please find attached the revised version.

Reviewer 2 Report
The reviewed manuscript covers important issues in the field of molecular biology that can be used in future biotechnological work on improving traits in crops. It should be emphasized that you have put a lot of work into a detailed analysis of important data and have developed very clear and legible figures that perfectly illustrate the enormity of your work. The figures mentioned are carefully designed to illustrate the relationships that you have studied. Due to the scientific and practical significance of your analyzes and problems formulated, I believe that the assessed work should be published unchanged.
Author Response
We thank you very much for your evaluation. Please note that the manuscript has been revised according to other reviewers.
Reviewer 3 Report
I.
I recommend reviewing the structure of the article.
Present in the following sequence:
1. Abstract
2. 1. Introduction
3. 2. Materials and methods
4. 3. Results and discusses
5. 4. Conclusions
6. Author Contributions
7. Funding
8. Institutional Review Board Statement
9. Data Availability Statement
1. Conflicts of Interest
1. Acknowledgments
1. References
II.
There are no conclusions.
Author Response
We thank you very much for helpful comment. As per your comments, we rearranged the structure of the article according to “International Journal of Molecular Sciences’ Instructions for Authors” and moved the conclusion of the results “section 2.3” into “Section 4. Conclusions” to clarify our conclusion. We have added “Section 4. Conclusions” in lines 226 in the revised manuscript.
Reviewer 4 Report
This paper is about a reliable research work related to a study of the identification of common 55 key residues for transferase activity in tr-GELPs using two-entropy analysis of amino acids, and the estimation of the effects of these residues on TciGLIP using structure model 57 prediction and docking simulations. The results presented in this paper are original and reflects a rigorous and serious work and team. Only that some minor revisions are needed as presented in the attached document.

Author Response
We thank this reviewer for evaluation and helpful comment. As per your comments, we revised and added spell out to your pointed words as follow:
In the Introduction section
“Gly-Asp-Ser-Leu (GDSL) motif esterase/lipase family proteins (GELPs) are lipases that feature a Gly-Asp-Ser-X (GDSX) consensus motif and are in-volved in a wide variety of biological functions [1], including seed germination [2], pollen interaction [3], lipid metabolism [4], and secondary metabolism [5].” (Line 31 in the revised manuscript.)
In the Results and Discussion section
“2.2. Molecular phylogenetic analysis of GELPs”
Interestingly, this cluster is located distantly from other Brassicaceae protein clusters, including est-GELP BnSCE3 (Brassica napus sinapine esterase) (Figure 3, clade D-II), suggesting that these GELPs might have been multiplied separately in the same plant family. (Line 121 in the revised manuscript.)
Reviewer 5 Report
The manuscript is well written and is very interesting.
In Line 67 replace serpentine with serpentina
I haven't find the conclusion section in the manuscript.
Author Response
We thank you very much for evaluation and helpful comment. As per your comments, we have moved the conclusion of the results “section 2.3” into “Section 4. Conclusions” to clarify our conclusion, and have added “Section 4. Conclusions” in Line 226 in the revised manuscript. In addeition, we have fixed “Rauvolfia serpentine acetylajmalan acetylesterase” to “Rauvolfia serpentina acetylajmalan acetylesterase” in Line 69 in the revised manuscript.
- I haven't find the conclusion section in the manuscript.
Round 2
Reviewer 3 Report
The article has only 25% of the analysis of publications for a 5-year retrospective. The relevance of the presented results has not been proven. The architecture of the article does not allow for a clear analysis of the presented scientific result.